# Evaluation of Growth Characteristics and Biological Activities of ‘Dachul’, a Hybrid Medicinal Plant of *Atractylodes macrocephala* × *Atractylodes japonica*, under Different Artificial Light Sources

**DOI:** 10.3390/plants11152035

**Published:** 2022-08-04

**Authors:** Myeong Ha Hwang, Ji Won Seo, Byung Jun Park, Kyeong Jae Han, Jae Geun Lee, Na Young Kim, Myong Jo Kim, Eun Soo Seong

**Affiliations:** 1Interdisciplinary Program in Smart Science, Kangwon National University, Chuncheon 24341, Korea; 2Division of Bioresource Sciences, Kangwon National University, Chuncheon 24341, Korea; 3Department of Herb Crop Resources, NIHHS, RDA, Eumseong 369-873, Korea; 4Research Institute of Biotechnology, Hwajinbiocosmetic, Chuncheon 24232, Korea; 5Hotel Culinary Arts, Songho University, Hoengseong 25242, Korea

**Keywords:** antioxidant activity, anti-inflammation activity, artificial light source, biological activities, Dachul, growth characteristics

## Abstract

This study was conducted to evaluate the effects of different artificial light sources on the growth characteristics and various biological activities of the *Atractylodes macrocephala* x *Atractylodes japonica* hybrid cv. ‘Dachul’, which is highly useful for medicinal purposes. The plant had the largest biomass with a plant height of 38.20 ± 1.95 cm when treated with microwave electrodeless light (MEL). The chlorophyll content of the plants treated with fluorescent light (FL) was 53.93 ± 1.05 SPAD and was the highest. The antioxidant effect, determined using 2,2-diphenyl-1-picrylhydrazyl (DPPH), was the highest with 92.7 ± 0.2% in plants treated with light-emitting diode (LED)-green light. Total phenol and flavonoid contents were significantly higher with 19.7 ± 0.5 mg GAE/g and 40.2 ± 2.2 mg QE/g in the sample treated with LED-green light, respectively. For antimicrobial activity using the minimum inhibitory concentration (MIC) technique, the inhibitory ability against Escherichia coli was at 0.25 mg/mL under LED-green light treatment. The whitening activity using tyrosinase enzyme showed the highest tyrosinase inhibitory ability at 62.1 ± 1.2% of the above extract treated with MEL light. To confirm the immune activity in lipopolysaccharide (LPS)-induced RAW 264.7 cells, NO production of inflammation-related substances was measured. In addition, the inflammation-related genes iNOS (inducible nitric oxide synthase), COX-2 (cyclooxygenase-2), and TNF-α (tumor necrosis factor-α) in the same sample were confirmed using reverse transcriptase (RT)-PCR, and the result showed that gene expression was suppressed compared with that in the control group. It is expected that Dachul plants treated with LED-blue light will play an important role in enhancing intracellular anti-inflammatory activity. From these results, the effect for various biological activities appeared in a significantly diverse spectrum in response to different wavelengths of artificial light sources in Dachul.

## 1. Introduction

*Atractylodes* spp. is a herbaceous perennial plant of the genus Asteraceae and is widely used in East Asian countries, such as Korea, Japan, and China, for medicinal purposes. There are five species in the genera: *A. lancea*, *A. chinensis*, *A. ovate*, *A. japonica*, and *A. macrocephala* [1]. The roots of *A. macrocephala* and *A. japonica* contain 1.5% volatile essential oil, and the main components include atractylone, atractylenolide (I, II, III), 3β-hydroxyatractylone, and 3β-acetoxyatractylone [2]. The main effects of A. japonica are reported to be diuresis, sweating, promotion of liver tissue regeneration, analgesic, antifungal, and anti-inflammatory effects [3]. Recently, the effects of enhancing biological and anti-inflammatory activities were revealed in a study on *A. macrocephala* using an artificial light source [4]. A. japonica is a species native to Korea and was developed by hybridizing *A. japonica* × *A. macrocephala* to overcome the problems of low seed rate, slow root growth, and low production. The multi-flowers are purple, and the bulbs have characteristics similar to those of *A. macrocephala*. The yield is higher than that of the native species, and the medicinal properties are also superior to those of the native species [5].

Recently, research on technologies related to production systems that can stably produce high-quality agricultural products has been increasing owing to abnormal weather, food shortages due to natural disasters, decreased agricultural production, and increased interest in food safety. As plant factories are reported to be alternatives that will overcome future food crises owing to abnormal weather, research in various fields related to plant factories is being actively conducted. Artificial light sources that can be used in plant factories include high-pressure sodium lamps, metal halide lamps, fluorescent lamps, and light-emitting diodes (LEDs). Since its development in the United States in the early 1960s, LED has not attracted much attention. However, in the 1990s, as high-brightness blue LEDs were developed in Japan, their applicability as an artificial light source for plant cultivation began to be actively studied [6]. LED is safe and environmentally friendly because it is mercury-free, and it has a longer lifespan compared to discharge lamps, including fluorescent lamps. In addition, LED consumes less power and is easier to select light quality (wavelength) and control the amount of light compared to other light sources. Moreover, its pulse irradiation is beneficial for photosynthesis. Many studies have been conducted on artificial light sources for plant cultivation [7]. As the amount or quality of light differs according to the light source, the growth response of plants also shows a huge difference. Research on this topic has been conducted on various crops, including lettuce. In particular, studies on the production of high-quality agricultural products are being actively carried out to increase the content of functional ingredients of crops, such as beta-carotene and anthocyanin, by irradiating a specific light wavelength using LED, which has recently been attracting attention for promoting growth [8,9,10]. MEL (microwave electrodeless light, electrodeless lighting) is a light source obtained by charging a gas to generate plasma light. Since it does not use a separate electrode, it has an advantage in that it has a longer lifespan compared to a light source that needs to consider the lifespan of the electrode. It is also known as an artificial light source that can generate a spectrum most similar to sunlight because it has a high color rendering index [11].

The antioxidant activity of plant-based functional metabolism can further increase when cultivated using artificial light sources. Among antioxidants, polyphenol compounds, such as flavonoids, anthocyanins, tannins, catechins, isoflavones, lignans, and resveratrol, are widely distributed in the plant kingdom and are present in large amounts in fruits and leafy vegetables [12,13]. Many of the hydroxyl groups (-OH) present in polyphenols can easily bond with various compounds and thus have excellent antioxidant, anticancer, and anti-inflammatory effects [14,15]. Flavonoids are compounds that belong to polyphenols and are a generic term for phenolic compounds that have C6-C3-C6 as their basic skeleton. Flavonoids are yellow or pale yellow and are widely distributed in nature like polyphenols. They are present in almost all parts, such as leaves, flowers, fruits, stems, and roots, and are abundant in grains and fruits [16]. Flavonoids have high antioxidant activity by effectively removing reactive oxygen species and are known to have antiviral, anti-inflammatory, and anticancer effects like polyphenols [17,18].

The purpose of this study was to analyze the differences in growth and various biological activities of the aerial and underground parts of the Dachul cultivar under different types of artificial light sources. (1) The difference in the growth of Dachul plants was measured using plant height, leaf length, leaf width, number of leaves, and dry weight, and the effect of artificial light sources on chlorophyll content was investigated. (2) Various biological parameters, such as antioxidant activity analyzed using the DPPH method; total phenol and flavonoid contents; and antimicrobial, whitening, and anti-inflammatory activities were evaluated.

## 2. Results and Discussion

### 2.1. Characteristic Difference in Growth and Chlorophyll of Dachul Treated with Different Artificial Light Sources

The result of investigating the plant growth characteristics of Dachul using artificial light source treatments showed that plant height, leaf length, and leaf width, excluding the number of leaves, were the highest with 3.20 ± 1.95, 8.27 ± 0.64, and 6.87 ± 0.90 cm, respectively in the above-ground part of the stem treated with MEL. (Table 1 and Figure 1a). The number of leaves was 18.00 ± 1.63 ea under LED-blue treatment and was the highest compared with that under other artificial light source treatments. There was no statistically significant difference in dry weight for the above-ground part between the FL, LED-blue, MEL, and SL treatments. The root length of the underground part ranged from 11.43 ± 1.10 cm to 13.66 ± 1.84 cm in all the treatment groups, except for LED-green, indicating that there was no significant difference between them. However, the dry weight of the underground part in the MEL and SL treatment groups was 0.21 ± 0.06 g and 0.23 ± 0.06 g, respectively, showing the highest dry weight. The result of examining the effect of different types of light sources on the growth of lettuce three weeks after planting showed an order as follows: white-LED, followed by EEFL 2, and then EEFL 1; however, there was no statistical significance between the light sources [19]. The high dry weight in the MEL and SL treatment groups seems to be related to the effect of continuous spectrum wavelength according to the light quality. According to a report by Lee et al. (2016), the number of leaves harvested by windbreak based on light quality was the highest in the fluorescent lamp treatment group, with 19.7 ea higher than that in the LED treatment [20]. However, our results showed that LED-blue resulted in the highest leaf production. Hence, there is a large difference between plant varieties under different sources of artificial light.

The SPAD value of the difference in the chlorophyll content of the Dachul plant under artificial light source treatments showed that the FL treatment group had the highest chlorophyll content at 53.93, and the LED-green treatment group had the lowest chlorophyll content at 35.90 (Figure 1b). There are still only a few cases of differences in chlorophyll content according to the light quality of artificial light sources. The chlorophyll content of kale under the light source was 57.3–59.3, and there was no significant difference between the treatments [21]. It has been reported that light quality affects the chlorophyll synthesis of plants [22]. Lee et al. (2010) reported that chlorophyll content can be controlled by adjusting the LED light quality in lettuce [9]. A report also shows that the chlorophyll content in lettuce increases with blue light treatment [23]. However, in our study, LED-blue light showed a lower chlorophyll content than FL. Hence, several important factors other than LED-blue light are expected to act on the chlorophyll synthesis mechanism.

### 2.2. Effects of Artificial Light Treatment on Antioxidants, Total Polyphenols, and Total Flavonoids in Dachul Plants

The antioxidant effect of Dachul plants under artificial light source treatments was analyzed using the DPPH method. The result revealed that all conditions, except for the SL treatment, showed a concentration-dependent increase between the aerial and underground parts (Figure 2). The overall level of free radical inhibitory activity was determined for all samples. In the LED-green treatment group, both the aerial and underground parts showed the highest antioxidant activity at a sample concentration of 2000 µg/mL, with 92.7 ± 0.2% and 92.3 ± 0.3%, respectively. In addition, at a sample concentration of 1000 µg/mL, the antioxidant effect was high (53.4 ± 0.7%) in the LED-green treatment group, and the difference was statistically significant. In another study, when the DPPH scavenging ability of extracts of each cultivar of *A. japonica* was measured, the free radical scavenging ability of Sangchul was found to be the highest. Moreover, Dawon and Huchul showed a tendency to increase in a concentration dependent manner up to 250 µg/mL [24]. The antioxidant activity of smart farm-cultivated plants using artificial light is mainly concentrated in leafy vegetables, and the use of medicinal plants is extremely limited. As the free radical scavenging ability of Dachul increased with the increase in artificial light sources in this study, there is a need to expand the facilities for growing medicinal plants using artificial light sources.

It has been reported that the more polyphenol compounds in plants, the higher the anticancer, antibacterial, and antioxidant activities, and tissues are protected from free radicals in the human body when these plants are ingested [25]. Results of measuring the total phenol content of Dachul under artificial light source treatments showed that the aerial-part extract had the highest content of 19.7 ± 0.5 mg GAE/g under LED-green light, and the underground part had 14.58 ± 0.4 mg GAE/g in the SL treatment group (Figure 3). Except for the LED-green light, there was no significant difference between the aerial and underground extracts under the different artificial light sources. The total polyphenol content was higher under LEP treatment than that under LED treatment during perilla breeding using an artificial light source, and it was reported that it had the highest content, particularly 53.1 ± 0.5 mg GAE/g in the stem of perilla [26]. When kale was grown under an artificial light source, the polyphenol content under red light and polyphenol synthesis effect under the mixed light quality of blue + red light was significant. However, the effect of the difference in light intensity was not significant in the two light conditions [27]. The difference in the polyphenol content of plants under the artificial light source treatment is considered to have a significantly greater effect depending on the color of the light source rather than the light intensity, and it is judged to be due to the difference in the response of plant varieties to the artificial light source.

The artificial light source-treated Dachul showed the highest flavonoid content in both the aerial and underground parts when LED-green light was used. The flavonoid content in the aerial part was 40.2 ± 2.2 mg QE/g, whereas that in the underground part was 12.2 ± 0.5 mg QE/g (Figure 4). The flavonoid content pattern according to the artificial light source using medicinal plants shows a large difference in the content response to a specific wavelength depending on the plant species. The total flavonoid content of root extracts treated with LED red and blue light was found to be higher than that of root extracts treated with other light sources in an experiment using perilla artificial light sources [26]. In cultivating kale using an artificial light source, it was found that the effect of light intensity on flavonoid synthesis was significant in the blue-white and blue-red mixed-light irradiated groups compared with that in the fluorescent and red light treatment groups [27]. Unlike polyphenols, flavonoids were found to affect material synthesis via light intensity. Although we did not investigate the relationship between light intensity and flavonoid synthesis in this experiment, it was found that LED-green had the greatest effect on flavonoid synthesis of Dachul plants, and the effect was significantly different from the other light sources.

### 2.3. Antimicrobial Activity

Through the minimum growth inhibitory concentration (MIC) experiment, the microbial inhibitory activity of each of the above- and below-ground extracts of Dachul was confirmed for six strains (Table 2). The aerial part of Dachul was resistant to *E. coli* and *P. aeruginosa* under all light treatment conditions. A minimum inhibitory concentration of 1 mg/mL was recorded, and no microbial inhibitory activity was observed at a lower concentration. Furthermore, microbial inhibitory activity was confirmed for *E. coli* and *V. litoralis* in the underground parts of Dachul. Results showed that 1 mg/mL in fluorescent treatment, 0.5 mg/mL in LED-red light treatment, 0.5 mg/mL in LED-blue light treatment, 0.25 mg/mL in LED-green light treatment, 0.5 mg/mL in MEL treatment, and 0.5 mg/mL in the SL treatment were the minimum inhibitory concentration against *E. coli* strain. The highest inhibitory activity was in the LED-green light treatment (Table 2). In a study on the growth of *Salvia miltiorrhiza* using an artificial light source, the aerial extract showed no microbial inhibition; however, the root extract showed minimal inhibitory ability against six types of microorganisms (*S. aureus*, *V. litoralis*, *B. subtilis*, *E. coli*, *S. typhimurium*, and *P. aeruginosa*) [28]. However, there are limited reports on the antimicrobial effects of plants produced using artificial light sources. In this study, we found that the antimicrobial effect of LED-green treated Dachul was greater than that of the other light sources. Therefore, there is a need to study the antimicrobial effects of plant extracts according to the quality of the artificial light sources in the future.

### 2.4. Whitening Activity

Dachul treated with an artificial light source showed that tyrosinase inhibitory activity increased as the sample concentration increased. When treated with MEL light, the aerial part extract of Dachul showed the highest inhibitory activity with 62.1 ± 1.2% (Figure 5). In the underground extract of Dachul, the LED-blue light-treated sample showed the highest tyrosinase inhibitory ability at 43.1 ± 2.5%, and there was a significant difference. Studies on the mechanism of inhibition of melanin synthesis for specific wavelengths of artificial light sources have not been studied much. When perilla was treated with artificial light sources, the tyrosinase inhibitory activity of LEP was higher than that of other light sources; however, the aerial part of Dachul showed higher tyrosinase inhibitory activity than LEP in MEL treatment [26]. In the LEP-green light sample, which had high antioxidant activity, the tyrosinase inhibitory effect was somewhat lower than that in the MEL light treatment group. Studies have shown that tyrosinase inhibitory ability increases in the plant treatment group with a large accumulation of antioxidant activities. Therefore, there is a need to study the mechanism of tyrosinase inhibition under various conditions [29].

### 2.5. Inhibition of NO Production and Expression of Inflammation-Related Genes in LPS-Induced RAW 264.7 Cells

The cytotoxicity test of aerial and underground part extracts of Dachul under light sources using the MTT assay revealed that the above-ground part showed lower cytotoxicity than the underground part. Cell viability was confirmed to be more than 70% at this concentration (Figure 6a,b). Cytotoxicity of the underground extracts of Dachul revealed that the cell viability was relatively low in the LED-red and LED-green treated groups compared with that in other light sources. In addition, the cells used in the experiment showed a cell viability of 70% or more at a concentration of 100 µg/mL or less compared with the extract in all other aerial- and underground-light source treatment groups. Hence, inhibition of NO production for RAW 264.7 cells proceeded under the condition of 100 µg/mL or less concentration.

The result of measuring the inhibition of NO production rate, which is a cause of inflammation, showed that the values were represented in RAW 264.7 cells induced by LPS using extracts of the aerial and underground parts treated with the artificial light source. At all concentrations of 10, 50, and 100 µg/mL treated with LED-green light, the anti-inflammatory response was 67.45 ± 0.74%, 64.25 ± 2.58%, and 51.43 ± 4.83%, respectively, showing the highest statistically significant values (Figure 6c). In the underground extract of Dachul, the LED-blue light treatment group showed the highest anti-inflammatory response with 9.08 ± 1.13% at 100 µg/mL and 51.43 ± 0.68% and 27.97 ± 1.88% in the LED-green treatment group at 10 and 50 µg/mL, respectively, indicating relatively high anti-inflammatory activity compared with other light sources (Figure 6d). In these results, the correlation that the anti-inflammatory activity is equally high in the treatment group with high antioxidant activity is not established. The activity of removing the factors that cause inflammation is expected to act as an artificial light source with a specific wavelength for the anti-inflammatory response for each plant species. As an example like this, the anti-inflammatory activity of A. japonica extracts is mostly due to the use of plants grown in general cultivation. There are few studies on the anti-inflammatory response of A. japonica using smart-farm cultivation technology, such as artificial light sources. A previous study analyzed the tendency of LPS-treated BV2 cells to decrease NO production in a concentration-dependent manner [24]. In this study, NO production decreased in a concentration-dependent manner in the aerial and underground extracts. To date, there have been few studies on the anti-inflammatory activity in A. japonica. Hence, as NO plays a physiologically important role in diseases related to immune function, it is thought that A. japonica will play a positive role in improving immunity against diseases.

NO production, which is one of the causes of inflammation, was measured in the extracts treated with artificial light sources, and the result indicated that the underground extract of the LED-blue-treated group showed the highest NO production inhibitory effect. The expression levels of iNOS, COX-2, and TNF-α were assayed using RT-PCR. In an experiment based on the premise that actin is expressed in the same amount, it was found that the expression of all the three genes related to inflammation decreased with LED-blue treatment compared with that in the control (Figure 7). iNOS is synthesized by the inflammatory stimuli in macrophages to produce NO, which is known to cause inflammatory diseases [30]. Therefore, our results suggest that the artificial light source LED-blue-treated Dachul suppresses the iNOS gene expression and lowers the inflammatory response. COX-2 is activated by inflammatory stimuli, including tumor necrosis factor-α (TNF-α) and LPS, or by promoting mitosis, and the expression of COX-2 promotes inflammatory responses in various inflammatory models [31]. In this study, the expression of iNOS, COX-2, and TNF-α genes reduced when the extract was treated with LED blue light. Therefore, it is thought that the LED-blue treatment may suppress the inflammation-stimulating factors of various skin cells.

## 3. Materials and Methods

### 3.1. Plant Materials and Growth Environment Using Artificial Light Sources

Experimental materials were obtained from multi-year-old seedlings grown for one month after low-temperature treatment from Department of Herb Crop Resources, NIHHS, Rural Development Administration in South Korea. Multiple seedlings with an average plant height of 12.4 cm were uniformly selected, transferred to a pot with a diameter of 10 cm containing sterilized topsoil, and transplanted. Five individuals for each light condition were then placed in a culture room maintained at a temperature of 25 ± 1 °C, followed by light treatment for 16 h. The humidity in the culture room was about 60%. They were grown for two weeks under repeated 8-h dark cycles. Five types of artificial light sources were used in this experiment: fluorescent light (FL), LED red light, LED blue light, LED green light, microwave electrodeless light (MEL), and sun light (SL). The characteristics of each light source are shown in Table 3. The wavelength and light intensity of a specific light source (Table 3) were measured using an illuminometer (PG200N, UPRtek, Miaoli, Taiwan).

### 3.2. Plant Gowth Characteristics

Plant growth was measured immediately after the two-week light treatment period for each treatment group. Plant height, leaf length, leaf width, number of leaves, and dry weight of the aerial part were measured, and the total length and dry weight of the underground part were measured. Length and growth were measured in the “cm” unit, and weight was measured in the “g” unit.

### 3.3. Chlorophyll Content

To measure chlorophyll content, a significant correlation between the soil and plant analyzer development (SPAD) value and chlorophyll content was tested using a SPAD502plus (Konica Minolta Co. Ltd., Tokyo, Japan) chlorophyll meter [32]. The average SPAD values were measured three times using the upper-third leaf for each plant.

### 3.4. Extraction and Concentration of Plant Samples

After harvesting the cuttings grown for two weeks in the culture room, the aerial and underground parts were separated and dried for three days using a freeze dryer (OPR-FDB-5003 Freeze Dryer, OPERON Co. Ltd., Hwanggeum-ro, Korea). The dried aerial and underground parts were crushed, and extraction was performed for three days using 100% MeOH. The extract was then filtered through a filter paper (Tokyu Roshi Kaisha Ltd., Tokyo, Japan) and concentrated under reduced pressure at a temperature of 45 °C using a rotary evaporator (EYELA N-1000, Tokyo Rikakikai Co. Ltd., Tokyo, Japan).

### 3.5. Analysis of Antioxidant Activity

First, 100 µL of 0.15 mM 1,1-diphenyl-2-picrylhydrazyl (DPPH; Alfa Aesar Co. Ltd., Tewksbury, MA, USA) was added to 100 µL of each sample diluted to concentrations of 500, 1000, and 2000 µg/mL. Next, the reaction was carried out in the dark at ambient temperature for 30 min, and the absorbance was measured at 519 nm using a UV-vis spectrophotometer (Multiskan FC Microplate Photometer, Thermo Fisher Scientific Inc., Waltham, MA, USA) [33].

### 3.6. Analysis of Total Polyphenol Content

Folin-Ciocalteu’s phenol reagent (50 µL; Sigma-Aldrich Co. Ltd., St. Louis, MO, USA) was added to 100 µL of 1 mg/mL sample and reacted for 5 min, followed by adding 300 µL of 20% Na2CO3 (Junsei Chemicals Co. Ltd., Tokyo, Japan). After stabilization for 15 min, 1 mL of distilled water was added, and the absorbance was measured at a wavelength of 740 nm using a UV-vis spectrophotometer (Multiskan FC Microplate Photometer, Thermo Fisher Scientific Inc. Ltd., Waltham, MA, USA) [34].

### 3.7. Analysis of Total Flavonoid Content

To determine total flavonoid content, 100 µL of 10% aluminum nitrate (Yakuri Co. Ltd., Tokyo, Japan) and 100 µL of 1M potassium acetate (Mallinckrodt Co. Ltd., Japan) were added to 500 µL of 1 mg/mL sample and reacted for 40 min. The absorbance was measured at 414 nm using a UV-vis spectrophotometer (Thermo Fisher Scientific Inc., Waltham, MA, USA) [35].

### 3.8. Analysis of Antimicrobial Activity

The antimicrobial activity of Dachul extract was measured using a serial two-fold dilution method [36]. The six strains used in the experiment are shown in Table 4. After diluting the bacterial culture to 200-fold using LB medium, it was dispensed into each well of a 96-well microplate, starting from the first horizontal row of the plate until the concentration of the sample ranged from 1000 µg/mL to approximately 7.8 µg/mL. The solution was diluted twice and dispensed sequentially. After culturing in the dark for 24 h in an incubator set for each growth temperature, the degree of bacterial growth was visually observed, and the minimum inhibitory concentration (MIC) that inhibited bacterial growth in the sample was measured.

### 3.9. Analysis of Whitening Activity

Forty microliters of 125U mushroom tyrosinase (Sigma-Aldrich Co. Ltd., St. Louis, MO, USA) prepared with 67 mM phosphate-buffered saline (PBS; pH 7.4; T&I Co. Ltd., Gyeonggi, Korea) as a solvent and 120 µL of 10 mM 3,4-dihydroxy-L-phenylalanine (L-DOPA; Sigma-Aldrich Co. Ltd., St. Louis, MO, USA) was added to 40 µL of sample and reacted for 30 min in an incubator set at 37 °C. In the dark, absorbance was measured at 519 nm using a UV-vis spectrophotometer (Thermo Fisher Scientific Inc., Waltham, MA, USA).

### 3.10. Analysis of Cell Viability

RAW 264.7 cells were acquired from the Korean Cell Line Bank and used in the experiment. Cell culture was performed by adding 10% fetal bovine serum (Hyclone Laboratories Inc., Logan, UT, USA) and 1% penicillin (100 U/mL; Lonza Walkersville Inc., Walkersville, MD, USA) to Dulbecco’s Modified Eagle Medium (DMEM) at 37 °C in a 5% CO_2_ incubator (MCO-19AIC, Snayo Co. Ltd., Tokyo, Japan). RAW 264.7 cells were aliquoted in a 96-well plate at a density of 10^5^ cells/well, and then cultured at 37 °C in a 5% CO_2_ incubator (MCO-19AIC; Snayo Co. Ltd., Tokyo, Japan) for 24 h, after which the supernatant medium was removed. Each sample (100 µL) was treated with concentrations of 10, 100, and 200 µg/mL [37]. After 24 h, the medium was removed again, and the MTT reagent was diluted to a concentration of 500 µg/mL in the medium used for cell culture. After 4 h, the supernatant was removed, and DMSO (Duchefa Biochemie Inc., Haarlem, Netherlands USA) was added to 100 µL. After 20 min, the absorbance was measured at 620 nm using a UV-vis spectrophotometer (Thermo Fisher Scientific Inc., Waltham, MA, USA).

### 3.11. Measurement of NO Production Rate in LPS-Induced Cell

The NO production rate of Dachul extract was measured using the Griess reagent system [38]. RAW 264.7 cells were aliquoted in a 96-well plate at a density of 10^5^ cells/well and cultured at 37 °C in a 5% CO_2_ incubator (MCO-19AIC; Snayo Co. Ltd., Japan) for 24 h, after which the supernatant medium was removed. Next, 100 µL of each concentration (10, 50, 100 µg/mL) of the sample containing lipopolysaccharide (LPS) were treated until the ratio of LPS to sample was 1:1 and incubated for 24 h in an incubator. Subsequently, 50 µL of the supernatant medium was removed, and the remaining medium with 1% sulfanilamide (Sigma-Aldrich Co. Ltd., St. Louis, MO, USA) diluted in Griess reagent [5% phosphoric acid (Wako Chemicals Inc., Richmond, VA, USA) + 0.1% N- 50 µL of (1-naphthyl)-ethylenediamine dihydrochloride (Sigma-Aldrich Co. Ltd., USA)] was treated and reacted. After 20 min, the absorbance was measured at 519 nm using a UV-vis spectrophotometer (Thermo Fisher Scientific Inc., Waltham, MA, USA).

### 3.12. Analysis of Gene Expression Using RT-PCR

Total RNA was extracted from the experimental material performed in Section 3.11 (Extraction Kit: Cat. No. 17211, Intron Co. Ltd., Gyeonggi, Korea). cDNA was synthesized, and reverse transcription PCR (RT-PCR) was performed. The nucleotide sequences of the primers used in the experiment are shown in Table 5. RT-PCR was proceeded with an initial denaturation of 5 min at 94 °C, followed by 30 cycles at 94 °C for 1 min, 55 °C for 1 min, and 72 °C for 1 min, with a final 10 min extension step at 72 °C. After performing electrophoresis using 1% agarose gel (Duchefa Biochemie Co. Ltd., Haarlem, The Netherlands) to confirm and compare the expression levels of inflammation-related genes using the amount of each synthesized cDNA, the Gel Doc XR+ Gel Documentation System (Bio-Rad Inc., Hercules, CA, USA) was used to determine the difference in gene expression.

### 3.13. Statistical Processing

IBM SPSS Statistics v24 program (SPSS, International Business Machines Co. Ltd., Armonk, NY, USA) was used to test the significance of the data for each experiment, which was replicated three times, and the data were compared using Duncan’s Multiple Range Test (DMRT) at 0.05 level of significance (*p* < 0.05).

## 4. Conclusions

This study investigated the growth characteristics, chlorophyll content, antioxidant activity, total phenol and flavonoid content, antimicrobial activity, whitening activity, and anti-inflammatory activity of Dachul by artificial light source treatment. Plant growth was the best when treated with MEL light source, and the chlorophyll content was the highest under fluorescent light. Antioxidant activity, total phenol and flavonoid content were highest under LED-green light. As for the antimicrobial activity, the inhibitory activity for *E. coli* was the highest in Dachul treated with LED-green light. Tyrosinase inhibitory activity was highest when treated with MEL light. As a result of NO production and inflammation-related genes, which are the criteria for anti-inflammatory activity, the highest anti-inflammatory effect was shown in the LED-blue light treatment. Since the physiological and biochemical mechanisms among antioxidant, antimicrobial, whitening, and anti-inflammatory activities are different, it is expected that differences in biological factors that respond to wavelengths of artificial light source will vary per plant species.

## Figures and Tables

**Figure 1 plants-11-02035-f001:**
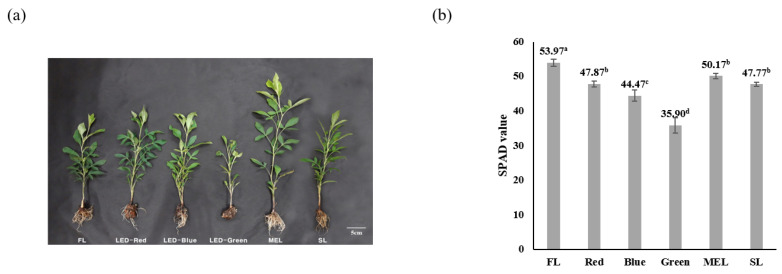
Growth characteristics (**a**) and chlorophyll contents (**b**) of Dachul under different light sources. Values represent mean of data obtained from three independent experiments (*p* < 0.05).

**Figure 2 plants-11-02035-f002:**
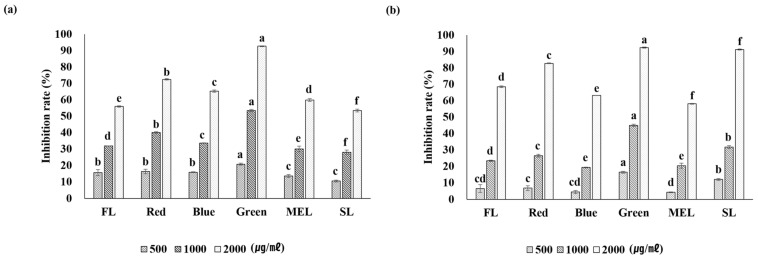
The inhibition rate of DPPH free radical scavenging activities of aerial (**a**) and underground (**b**) parts from extracts of Dachul under different light sources. Values represent mean of data obtained from three independent experiments (*p* < 0.05).

**Figure 3 plants-11-02035-f003:**
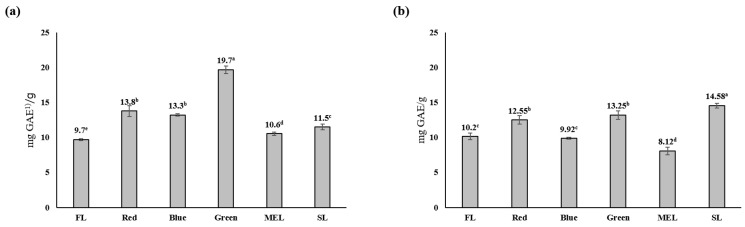
Total phenolic contents of aerial (**a**) and underground (**b**) parts from extracts of Dachul under different light sources. ^(1)^ GAE; gallic acid equivalents. Values represent mean of data obtained from three independent experiments (*p* < 0.05).

**Figure 4 plants-11-02035-f004:**
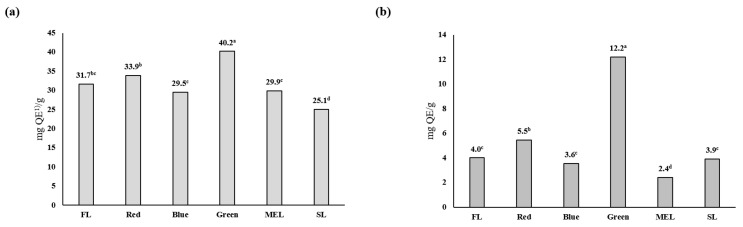
Total flavonoid contents of aerial (**a**) and underground (**b**) parts from extracts of Dachul under different light sources. ^(1)^ QE; quercetin equivalents. Values represent mean of data obtained from three independent experiments (*p* < 0.05).

**Figure 5 plants-11-02035-f005:**
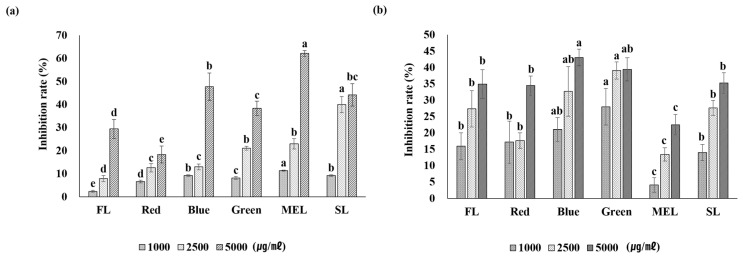
The inhibition rate of tyrosinase of aerial (**a**) and underground (**b**) parts from extracts of Dachul under different light sources.

**Figure 6 plants-11-02035-f006:**
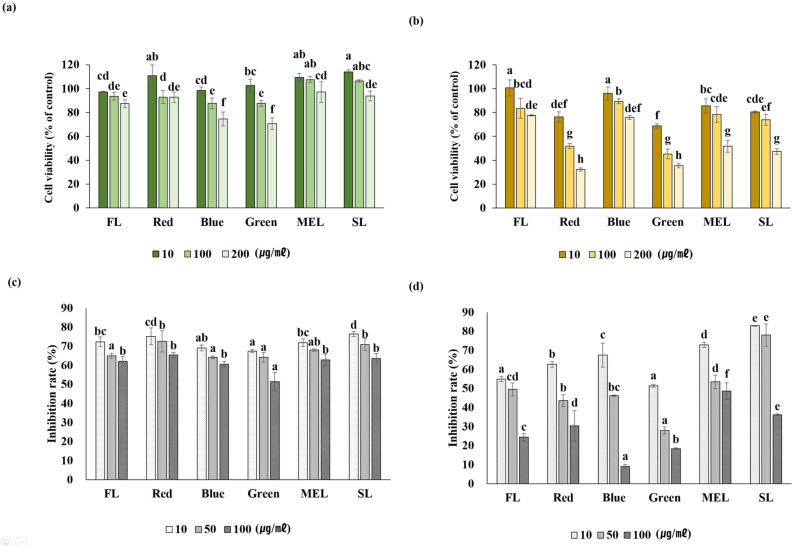
Cell viability (**a**,**b**) by MTT assay and NO production (**c**,**d**) in LPS-induced Raw 264.7 cell with Dachul extracts. (**a**,**c**); aerial part of Dachul, (**b**,**d**); underground part of Dachul.

**Figure 7 plants-11-02035-f007:**
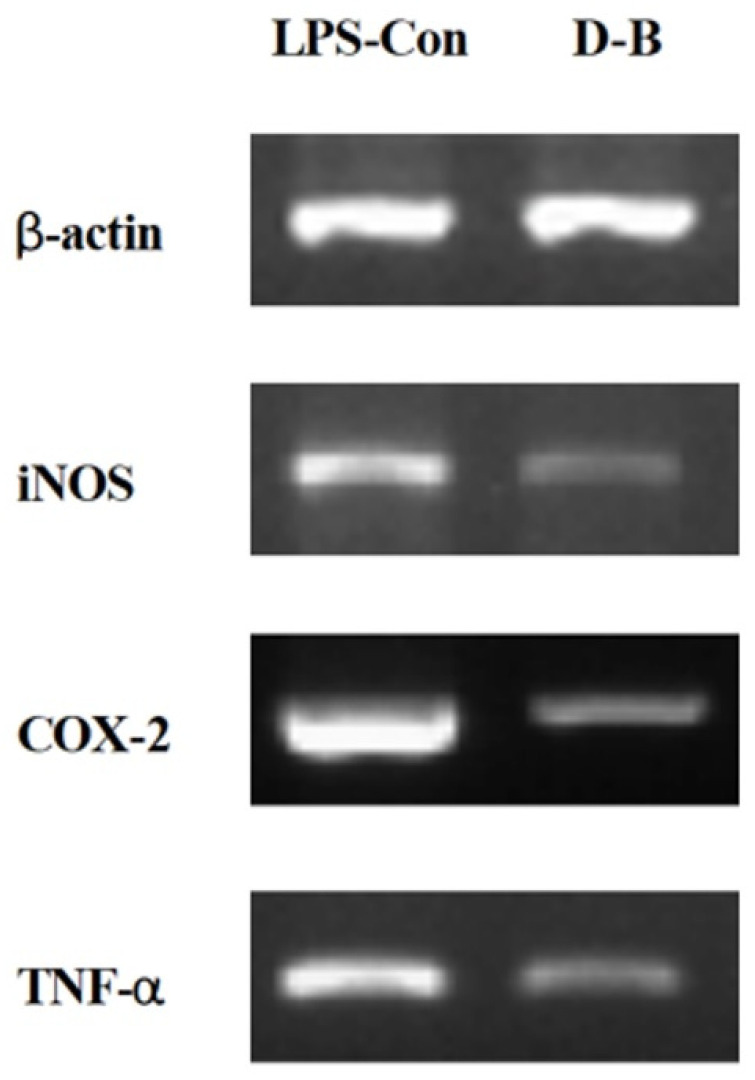
Transcriptional expression levels of iNOS, COX-2, and TNF-α genes in LPS-induced Raw 264.7 cells with underground parts extracts of Dachul. LPS-con; Control, D-B; Dachul grown under LED-Blue light.

**Table 1 plants-11-02035-t001:** Growth characteristics by parts of aerial and underground in Dachul treated with artificial light sources for 2 weeks.

Light Source	Aerial Part					Underground Part	
	Plant Length (cm)	Leaf Length (cm)	Leaf Width (cm)	Number of Leaves (ea)	Dry Weight (g)	Root Length (cm)	Dry Weight (g)
FL ^(1)^	24.47 ± 0.59 ^c^	6.03 ± 0.45 ^d^	5.20 ± 1.25 ^bc^	15.67 ± 0.58 ^a^	1.11 ± 0.06 ^ab^	12.13 ± 2.67 ^a^	0.19 ± 0.04 ^ab^
LED-Red	29.90 ± 3.06 ^b^	7.57 ± 0.64 ^ab^	6.23 ± 0.46 ^ab^	16.67 ± 1.53 ^a^	0.85 ± 0.06 ^b^	12.00 ± 1.20 ^a^	0.13 ± 0.01 ^bc^
LED-Blue	30.23 ± 1.94 ^b^	7.07 ± 0.51 ^bc^	5.57 ± 0.21 ^abc^	18.00 ± 1.63 ^a^	1.17 ± 0.24 ^ab^	13.66 ± 1.84 ^a^	0.18 ± 0.03 ^ab^
LED-Green	23.07 ± 1.56 ^c^	5.67 ± 0.06 ^d^	4.80 ± 0.00 ^c^	12.33 ± 0.58 ^b^	0.45 ± 0.18 ^c^	5.53 ± 0.57 ^b^	0.10 ± 0.01 ^c^
MEL ^(2)^	38.20 ± 1.95 ^a^	8.27 ± 0.64 ^a^	6.87 ± 0.90 ^a^	17.67 ± 1.53 ^a^	1.14 ± 0.06 ^ab^	11.43 ± 1.10 ^a^	0.21 ± 0.06 ^a^
SL ^(3)^	30.87 ± 3.06 ^b^	6.23 ± 0.50 ^cd^	4.97 ± 0.72 ^bc^	17.67 ± 2.31 ^a^	1.21 ± 0.01 ^a^	12.10 ± 2.00 ^a^	0.23 ± 0.06 ^a^

^(1)^ FL; fluorescent light, ^(2)^ MEL; microwave electrodeless light. ^(3)^ SL; sun light. Values represent mean ± S.D. of data obtained from three independent experiments. Duncan’s Multiple Range Test at 5% level (DMRT, *p* < 0.05).

**Table 2 plants-11-02035-t002:** Antimicrobial activity by parts of aerial and underground from extracts of Dachul treated with artificial light sources for 2 weeks.

Minimal Inhibitory Concentration (mg/mℓ)							
Dachul		*S. aureus*	*V. litoralis*	*B. subtilis*	*E. coli*	*S. typhimurium*	*P. aeruginosa*
Light Source	Plant Part						
FL	Aerial	ND ^(1)^	ND	ND	≥1	ND	≥1
	Underground	ND	≥1	ND	≥1	ND	ND
LED-Red	Aerial	ND	ND	ND	≥1	ND	≥1
	Underground	ND	≥0.5	ND	≥0.5	ND	ND
LED-Blue	Aerial	ND	ND	ND	≥1	ND	≥1
	Underground	ND	≥1	ND	≥0.5	ND	ND
LED-Green	Aerial	ND	ND	ND	≥1	ND	≥1
	Underground	ND	≥0.25	ND	≥0.25	ND	ND
MEL	Aerial	ND	ND	ND	≥1	ND	≥1
	Underground	ND	≥0.5	ND	≥0.5	ND	ND
SL	Aerial	ND	ND	ND	≥1	ND	≥1
	Underground	ND	≥0.5	ND	≥0.5	ND	ND
Tetracycline		≥0.007	≥0.007	≥0.007	≥0.007	≥0.007	≥0.007

The MIC values against bacteria were determined by the serial 2-fold dilution method. ^(1)^ ND; not detected.

**Table 3 plants-11-02035-t003:** The characteristics of artificial light sources used in the experiment.

Type	Wavelength	Light Intensity
Fluorescent light	continuous spectrum	24 µmol/m^2^·s
LED-Red	630 nm	22 µmol/m^2^·s
LED-Blue	450 nm	14 µmol/m^2^·s
LED-Green	520 nm	12 µmol/m^2^·s
Microwave electrodeless light	continuous spectrum	52 µmol/m^2^·s
Sun light	continuous spectrum	2.8 µmol/m^2^·s

**Table 4 plants-11-02035-t004:** List of microorganism strains and growth conditions used in antimicrobial activity.

Microorganism		Media	Incubation Temperature
*Bacillus subtilis*	KCTC 1021	Nutrient	30 °C
*Staphylococcus aureus*	KCTC 1916	Nutrient	37 °C
*Escherichia coli*	KCTC 1924	Nutrient	37 °C
*Salmonella typhimurium*	KCTC 1925	Nutrient	37 °C
*Pseudomonas aeruginosa*	KCTC 2742	Nutrient	37 °C
*Vibrio litoralis*	KCTC 13228	Nutrient	37 °C

**Table 5 plants-11-02035-t005:** Gene annotations and primer sequences used in RT-PCR analysis.

Primer		Primer Sequence (5′ → 3′)
β-actin	Forward Reverse	TGACGGGGTCACCCACACTGTGCCCATCTA CTAGAAGCATTTGCGGTGGACGATGGAGGG
iNOS	Forward Reverse	CCCTTCCGAAGTTTCTGGCAGCAGC GGCTGTCAGAGCCTCGTGGCTTTGG
COX-2	Forward Reverse	CACTACATCCTGACCCACTT ATGCTCCTGCTTGAGTATGT
TNF-α	Forward Reverse	TTGACCTCAGCGCTGAGTTG CCTGTAGCCCACGTCGTAGC

## Data Availability

The data presented in this study are contained within the article.

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
