# Peer review of "Evaluation of Growth Characteristics and Biological Activities of ‘Dachul’, a Hybrid Medicinal Plant of *Atractylodes macrocephala* × *Atractylodes japonica*, under Different Artificial Light Sources"

_plants, 2022, doi:10.3390/plants11152035_

Round 1
Reviewer 1 Report
I don't have any recommendations
Author Response
Response to Reviewer 1 Comments
As there were no specific points pointed out, there is no revision for Reviewer 1.
Reviewer 2 Report
It is an interesting and current study on the influence of the type of artificial light on the growth of plants and bioactivity of plant extracts. The main strength of the study is the large number of analyzes performed.
My specific recommendations are:
Some information about microwave electrodeless light, a lesser known type of light compared to the rest, should be included in the introduction.
Figure 2, 5 and 6 are unclear, please change it and possibly increase the font.
For the method, please specify which culture chamber was used, what was the humidity, how the watering was done and what size the pots were. Please specify how you treated the plants in the sunlight, they were probably exposed to direct sunlight. Hasn't the intensity been measured for sunlight? If it was measured please enter the value in table 3. Please specify where the lights were purchased.
I think the abstract could be improved.
In terms of results and discussion, the presentation of discussions based on the literature should be made more cursive. In this form the discussion is abrupt. There is a need for continuity between the results obtained in this study and those in the literature.
Author Response
Response to Reviewer 2 Comments
Most of the corrections pointed out by reviewer 2 are marked in red.
Point 1: Some information about microwave electrodeless light, a lesser known type of light compared to the rest, should be included in the introduction.
Response 1: Revised by adding to the introduction.
Point 2: Figure 2, 5 and 6 are unclear, please change it and possibly increase the font.
Response 2: I have re-edited the picture by enlarging the font.
Point 3: For the method, please specify which culture chamber was used, what was the humidity, how the watering was done and what size the pots were. Please specify how you treated the plants in the sunlight, they were probably exposed to direct sunlight. Hasn't the intensity been measured for sunlight? If it was measured please enter the value in table 3. Please specify where the lights were purchased.
Response 3: It was described in detail in 3.1 of Materials and Methods, and modified by adding it to Table 3.
Point 4: I think the abstract could be improved.
Response 4: The reviewers did not point out in detail about the abstract. I think the current state is sufficient.
Point 5: In terms of results and discussion, the presentation of discussions based on the literature should be made more cursive. In this form the discussion is abrupt. There is a need for continuity between the results obtained in this study and those in the literature.
Response 5: It has been modified by attaching a sentence that flexibly connects the contents of the result and discussion.

Round 2
Reviewer 2 Report
I think the author has fulfilled the requirements and I have no other specific recommendations.
The light intensity (Table 3) seems low, especially in sunlight, I am personally curious how the intensity was measured. This aspect could also be mentioned in the paper (method of measuring light intensity).
Author Response
Response to Reviewer 2(2nd)
Point 1: I think the author has fulfilled the requirements and I have no other specific recommendations.
The light intensity (Table 3) seems low, especially in sunlight, I am personally curious how the intensity was measured. This aspect could also be mentioned in the paper (method of measuring light intensity).
Response 1: I revised by referring in Materials and Methods 3.1. It is marked in blue.
Also, It has been modified in response to comments (introduction revision) from the Academic editor.
